# Regulatory-Compliant Infrastructure for AI Validation and Lifecycle Management in Healthcare

**Abstract.** This paper presents a robust, regulatory-compliant infrastructure specifically developed to address the validation and lifecycle management of artificial intelligence (AI) applications in healthcare. This infrastructure enables rigorous validation, seamless integration, and continuous monitoring of AI-driven healthcare solutions in alignment with established regulatory guidelines. By emphasizing transparency, reproducibility, and interoperability, the proposed infrastructure facilitates trust and adoption among stakeholders. Key components include curated public and proprietary datasets, standardized validation workflows, structured Data Use Agreements (DUAs), comprehensive version control, defined access rights, data sequestration protocols, traceability, audit trails, and anti-competitive safeguards within a multi-stakeholder consortium comprising data providers, data users, model providers, model users, and technology and service providers.

**Keywords:** AI validation, lifecycle management, regulatory compliance, Good Machine Learning Practice, dataset integration, continuous monitoring, audit trails, federated learning, data governance, multi-stakeholder consortium

## 1    Introduction

The rapid integration of AI in healthcare has introduced notable opportunities alongside significant regulatory and patient safety challenges. Regulatory bodies worldwide emphasize the need for rigorous validation and lifecycle management to ensure AI systems are safe, effective, and equitable. Current practices often lack standardized and comprehensive frameworks, risking inconsistent outcomes and regulatory non-compliance (U.S. FDA, 2025; Shah & Halamka, 2023–2024). This paper introduces a structured and compliant infrastructure, supported by a diverse stakeholder consortium, to systematically validate, deploy, and continuously monitor healthcare AI solutions, ensuring they meet or exceed regulatory standards.

## 2    Background

AI in healthcare must navigate stringent regulatory landscapes designed to protect patient safety and uphold efficacy. Regulatory agencies like the FDA mandate Good Machine Learning Practice (GMLP) to manage risks associated with AI systems (U.S. FDA, 2025). Despite these guidelines, healthcare organizations frequently encounter challenges such as limited access to validated datasets, inconsistent validation methods, and insufficient continuous monitoring practices (Maslej, 2025; Lin, 2025). These challenges hinder widespread adoption and integration of AI solutions.

# 3 Methodology

The proposed infrastructure addresses these gaps through three core components:

## 3.1 Dataset Integration

We leverage both publicly available and proprietary datasets provided by healthcare organizations and third-party data aggregators. Structured and regulatory-compliant Data Use Agreements (DUAs) govern dataset integration. Data annotation processes utilize pre-annotated data provided by AI model providers, augmented and validated by board-certified physician experts, such as radiologists (Jimenez-Pastor et al., 2023; Ji et al., 2025). Dataset management includes comprehensive version control, rigorous data sequestration protocols distinguishing between training, testing, and validation datasets, and defined access rights tailored to various user roles (e.g., data scientists vs. clinicians). Figure 2 illustrates detailed steps involved, including data acquisition, annotation, governance, versioning, and access control processes, emphasizing segregation and traceability throughout the dataset lifecycle.

## 3.2 Validation Framework

Our infrastructure provides standardized validation protocols aligned with regulatory guidelines, specifically Good Machine Learning Practice (GMLP). Validation methods encompass quantitative performance assessments, robustness testing across clinical scenarios, fairness and bias evaluation, and generalizability analysis (Pati et al., 2022). Structured workflows facilitate reproducible experiments, incorporating expert reviews, controlled benchmarking, and automated reporting mechanisms. Figure 3 outlines a comprehensive validation process, highlighting systematic checkpoints and iterative validation cycles, ensuring consistency, transparency, and reproducibility (Bracci et al., 2024).

## 3.3 Lifecycle Management

Continuous monitoring ensures AI systems maintain effectiveness and compliance post-deployment. Real-time performance tracking is coupled with automated alerts for performance degradation or compliance deviations. Periodic revalidation protocols include retrospective analyses and scenario-based stress tests to ensure sustained reliability and regulatory alignment over time. Comprehensive audit trails record all user activities, ensuring full traceability and integrity of both models and datasets, particularly critical for updates following the Predetermined Change Control Plan (PCCP) (Beutel et al., 2022; Ma et al., 2024). Figure 4 depicts the lifecycle management protocol, clearly demonstrating processes for continuous monitoring, audit trails maintenance, and systematic revalidation activities.

# 4 Implementation

The implementation of the infrastructure involves two main phases:

## 4.1 Initial Phase

- Establishing robust stakeholder partnerships within a multi-stakeholder consortium, including data providers, data users, model providers, model users, regulatory bodies, and technology and service providers (Tizhoosh, 2025).
- Designing subsidy models for under-resourced data providers to promote inclusive participation (Shah & Halamka, 2023–2024).
- Defining clear, standardized, non-exclusive Data Use Agreements (DUAs) and anti-competitive measures to ensure fair access and broad participation.

## 4.2 Subsequent Phases

- Expanding the dataset ecosystem by integrating additional high-quality public and proprietary datasets (Voss et al., 2015).
- Comprehensive integration with stakeholders to facilitate widespread adoption (Barreto et al., 2012; Christen, 2012).
- Ongoing regulatory alignment through continuous engagement and iterative enhancements to validation and monitoring protocols (U.S. FDA, 2025).

# 5 Results

Preliminary implementation outcomes include:

- Formation of an effective multi-stakeholder consortium comprising data providers, data users, model providers, model users, and technology and service providers (Tizhoosh, 2025).
- Defined and agreed-upon standardized validation and lifecycle management processes (Bracci et al., 2024).
- Successful demonstration of initial infrastructure capabilities through pilot validation studies, indicating improvements in validation consistency and regulatory alignment (Samanta et al., 2022).

# 6 Discussion

The presented infrastructure significantly mitigates patient safety and regulatory compliance risks by providing standardized, robust validation and monitoring practices. By addressing fundamental barriers such as dataset access, validation reproducibility, and continuous monitoring, this approach represents a scalable and sustainable model for healthcare AI adoption (Adnan et al., 2022).

Potential limitations include variability in dataset quality and availability, evolving regulatory standards requiring continuous adaptation, and resource-intensive validation processes. These challenges highlight the importance of ongoing refinement, automation of workflows, and further integration with emerging regulatory guidance (Qin et al., 2025). Comparisons with existing solutions underscore the comprehensive and systematic nature of our approach, emphasizing enhanced reproducibility, transparency, and interoperability (Koutsoubis et al., 2024).

Future enhancements will focus on further automating validation workflows, expanding dataset diversity through additional partnerships, and refining continuous monitoring algorithms to proactively identify emerging performance trends and compliance needs (Guo et al., 2024).

## 7    Diagrams

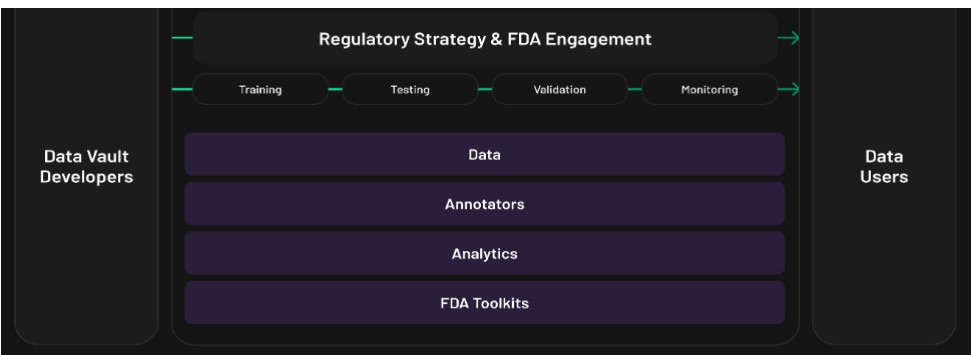

**Fig. 1.** Infrastructure Overview

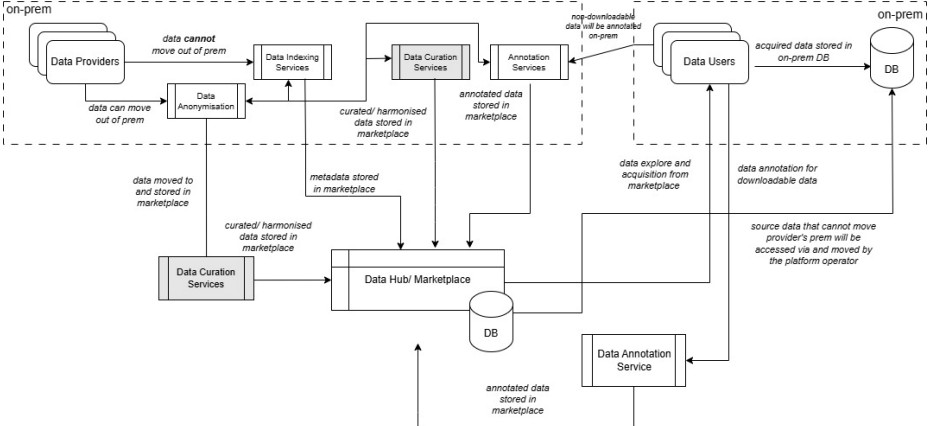

**Fig. 2.** Dataset Integration Workflow - Illustrating detailed steps from data acquisition, version control, annotation governance, and access control

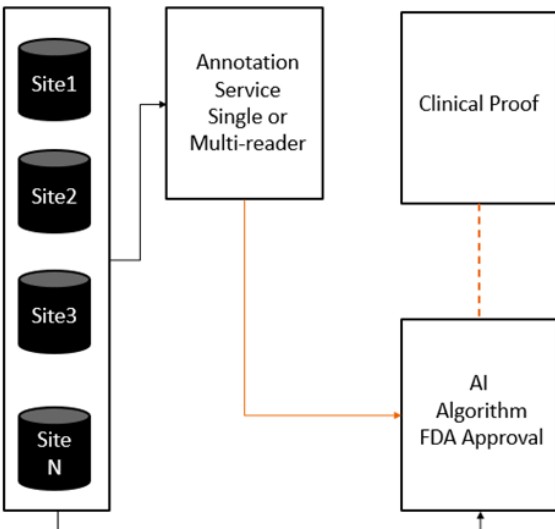

**Fig. 3.** Validation Process Flowchart - Showing systematic validation steps

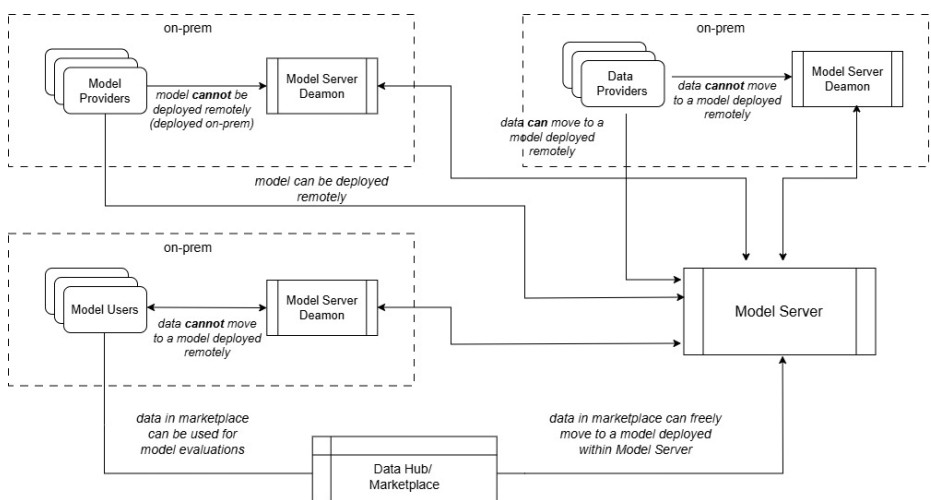

**Fig. 4.** Lifecycle Management Protocol

## 8 Conclusion

We propose a comprehensive, regulatory-compliant infrastructure that systematically addresses validation and lifecycle management challenges in healthcare AI. Supported by a robust multi-stakeholder consortium, this infrastructure facilitates widespread adoption, regulatory compliance, and improved patient safety, establishing a foundational model for responsible and scalable AI deployment within healthcare settings.

Additionally, this infrastructure can serve as a coordinating and reconciling platform for emerging certification groups such as CHAI, Joint Commission, URAC, and NCQA.

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
