# OpenReview forum: "Regulatory-Compliant Infrastructure for AI Validation and Lifecycle Management in Healthcare"
_MICCAI.org/2025/Workshop/BRIDGE — BRIDGE 2025 Poster_

### Official Review · Reviewer_ZLLt · 2025-07-24
**The paper  presents a nice  regulatory-compliant infrastructure, but  lacks deeper analysis and discussion of practical challenges**

**Rating:** 6
**Confidence:** 4

**Review:**

This paper proposes a regulatory-compliant infrastructure developed to address the validation and lifecycle management ofAI applications in healthcare. The authors present a three-component framework encompassing dataset integration, standardized validation workflows, and continuous lifecycle management, supported by a multi-stakeholder. The infrastructure aims to ensure AI systems meet regulatory standards like FDA's Good Machine Learning Practice while facilitating widespread adoption through standardized processes and comprehensive audit trails.

Paper strengths:
1. Tackles the genuine gap in standardized, regulatory-compliant validation frameworks for healthcare AI deployment/
2.  Recognizes diverse stakeholder needs and proposes inclusive governance structure with data providers, model developers, healthcare organizations, and regulators
3. The paper is very well aligned with BRIDGE.
Paper limitations:
1. Missing discussion of specific next steps, pilot implementation strategies, or roadmap for scaling the proposed infrastructure
2. For example, while the paper acknowledges challenges like "dataset quality variability" and "resource-intensive processes", it doesn't propose actionable solutions or mitigation strategies.

We recommend adding one paragraph discussing concrete approaches to address identified barriers.

---

### Official Review · Reviewer_nGf1 · 2025-07-25
**The proposal is ambitious but lacks key details that address practical implementation challenges.**

**Rating:** 6
**Confidence:** 4

**Review:**

## Summary of the Paper
This paper proposes a comprehensive infrastructure framework designed to address validation and lifecycle management challenges for AI applications in healthcare through a multi-stakeholder consortium approach. The authors present three core components: dataset integration with structured Data Use Agreements, standardized validation protocols aligned with Good Machine Learning Practice, and continuous monitoring systems for post-deployment compliance. The work aims to establish a foundational approach for regulatory-compliant AI deployment in healthcare settings.

## Strengths
- Addresses Critical Healthcare AI Gap: The paper tackles a genuine and pressing need in healthcare AI - the lack of standardized, regulatory-compliant validation and lifecycle management frameworks that currently hinder widespread adoption.
- Comprehensive Systems Approach: The proposed infrastructure thoughtfully integrates multiple essential components including dataset governance, validation protocols, continuous monitoring, and audit trails within a unified framework rather than addressing these elements in isolation.
- Multi-Stakeholder Consortium Model: The inclusion of diverse stakeholders (data providers, model providers, regulatory bodies, technology providers) represents a realistic approach to addressing the complex ecosystem requirements for healthcare AI deployment.

## Limitations or Areas for Improvement
- Lack of Concrete Implementation Evidence: While the paper claims "preliminary implementation outcomes" and "successful demonstration of initial infrastructure capabilities through pilot validation studies," no specific results, metrics, performance data, or case studies are provided to substantiate these claims.
- Insufficient Stakeholder Consortium Details: The multi-stakeholder consortium is central to the proposal but lacks critical operational details including governance structure, decision-making processes, conflict resolution mechanisms, funding models, and specific roles/responsibilities of each stakeholder type.
- Unaddressed Stakeholder Incentive Misalignment: The authors' proposed multi-stakeholder consortium model has been discussed in previous publications. Although this approach may work, the limitations of mismatch in incentives between different stakeholders and how it hampers the overall goal of the paper and how to mitigate these challenges is not discussed.
- Missing Comparative Analysis: The paper lacks comparison with existing validation frameworks, regulatory pathways, or infrastructure solutions, making it difficult to assess the novelty and advantages of the proposed approach.

## Relevance to BRIDGE Workshop Topics
- Robust evaluation methods and regulatory frameworks for AI-enabled medical devices
- Post-market monitoring strategies to ensure ongoing safety and effectiveness
- Frameworks for monitoring AI performance in the real world
- Best practices for deploying continual learning systems under regulatory constraints

---

### Official Review · Reviewer_jroY · 2025-07-26
**Review Comments**

**Rating:** 6
**Confidence:** 4

**Review:**

This paper presents a framework for regulatory-compliant infrastructure designed to address AI validation and lifecycle management challenges in healthcare. The authors propose a three-component approach for dataset integration, standardized validation protocols, and continuous lifecycle management.
Paper Strengths: The paper tackles an important problem in healthcare AI as there is currently a challenge in transitioning AI research into medical products given the lack of understanding of AI regulations and the limited access to infrastructure to facilitate this transition. The three-component structure (dataset integration, validation framework, lifecycle management) is logically organized and covers key aspects of the AI development and deployment pipeline. However, there are concerns that need to be addressed.
The paper presents "comprehensive, regulatory-compliant infrastructure" but provides insufficient detail about how this infrastructure actually works, where it is located, how stakeholders can access it, or what specific technologies underpin it. Readers need concrete information about:
Is this a web-based platform? A software framework? A set of protocols?
How can organizations access and use this infrastructure?
What are the technical specifications and requirements?
The paper references Figures 1-4 throughout the text. Please enhance the resolution of the figures.
While the authors nicely summarized the data challenges, it is important to note that data curation for regulatory compliance involves numerous activities such as:
Data artifact detection and flagging
Outlier identification and management
Diversity and representativeness measurement
Coverage assessment of intended use populations
Bias detection and mitigation strategies
Data provenance and lineage tracking
Please expand the data section to discuss these issues and cite relevant works. See the two examples below, and how your framework is different than them.
There are key elements missing in the validation stage. For example:
FDA mandates subgroup analysis for all relevant confounding factors and demographic groups within the intended use population. The paper should clarify whether this is incorporated into the framework or represents a current limitation.
Specific methodologies for performance assessment, confidence intervals, and statistical significance testing.
Please discuss if these are part of your framework, and if not, list them as current limitations.
Figure 4 should clearly illustrate how post-market monitoring integrates with the overall framework. Additionally, please better distinguish between post-market monitoring and PCCP; these are related but distinct regulatory concepts that require clarification.
The paper needs stronger motivation for each framework component. For each stage, the authors should:
Clearly articulate the current gap in existing infrastructure or practices
Explain how their framework addresses this gap
Provide evidence or examples of why current approaches are insufficient
Demonstrate value proposition for different stakeholder groups
The paper is currently 7 pages with a 10-page limit, providing the opportunity for substantial improvement. Please expand the paper by discussing the issues listed above. Again, this is an important work that nicely summarizes the challenges and need for infrastructure to translate AI research into practice, and with the proposed improvements, it can lead to a meaningful impact.

---

### Decision · Program_Chairs · 2025-07-26

**Decision:**

Accept (Poster)

**Comment:**

Dear Authors,

We are pleased to inform you that your paper has been accepted for publication, conditional on addressing the reviewer comments below. Please revise accordingly and submit your updated manuscript by July 31. This work presents a needed infrastructure for translating AI research into products. We look forward to your improvements.

Requirements for your final camera‑ready submission (due July 31):
* Incorporate reviewer comments and suggestions throughout your paper; at minimum, add a discussion section that acknowledges and responds to the key points raised by reviewers.
* Ensure your final draft adheres to Springer formatting guidelines.
* Submit your camera‑ready source file along with any supplementary material.

Best regards,
BRIDGE Workshop